# SHOULD WE BE *Pre*-TRAINING ? EXPLORING END-TASK AWARE TRAINING IN LIEU OF CONTINUED PRE-TRAINING

**Lucio M. Dery**[1]**, Paul Michel**[2]**, Ameet Talwalkar**[1,3] **& Graham Neubig**[1]
[1] Carnegie Mellon University, [2] ENS PSL University, [3] Determined AI

ldery@andrew.cmu.edu, pmichel31415@gmail.com, talwalkar@cmu.edu, gneubig@cs.cmu.edu

## ABSTRACT

In most settings of practical concern, machine learning practitioners know in advance what end-task they wish to boost with auxiliary tasks. However, widely used methods for leveraging auxiliary data like pre-training and its continued-pretraining variant are end-task agnostic: they rarely, if ever, exploit knowledge of the target task. We study replacing end-task agnostic continued training of pre-trained language models with end-task aware training of said models. We argue that for sufficiently important end-tasks, the benefits of leveraging auxiliary data in a task-aware fashion can justify forgoing the traditional approach of obtaining generic, end-task agnostic representations as with (continued) pre-training. On three different low-resource NLP tasks from two domains, we demonstrate that multi-tasking the end-task and auxiliary objectives results in significantly better downstream task performance than the widely-used task-agnostic continued pre-training paradigm of Gururangan et al. (2020). We next introduce an online meta-learning algorithm that learns a set of multi-task weights to better balance among our multiple auxiliary objectives, achieving further improvements on end-task performance and data efficiency.

## 1 INTRODUCTION

The increasingly popular pre-training paradigm (Dai & Le, 2015; Devlin et al., 2018; Gururangan et al., 2020) involves first training a *generalist* model on copious amounts of easy-to-obtain data, e.g. raw text data in NLP, and then using this model to initialize training on a wide swath of downstream tasks. Generalist models like BERT (Devlin et al., 2018), RoBERTa (Liu et al., 2019), and GPT-3 (Brown et al., 2020) have a strong appeal; a few institutions with significant resources incur the cost of training these large models whilst the rest of the research community enjoys a significant performance improvement at minimal computational overhead. However, the advantages of initializing a downstream task from a generalist model are not guaranteed. Previous work has shown that the benefits of pre-training depend heavily on the degree of domain overlap between the end-task data and the massive, heterogenous data on which the generalist model was trained (Beltagy et al., 2019; Gururangan et al., 2020).

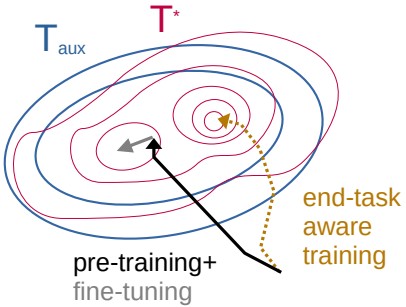

Figure 1: Pre-training trains on auxiliary task $T_{\text{aux}}$ before fine-tuning on primary task $T^*$. End-task aware training optimizes both $T_{\text{aux}}$ and $T^*$ simultaneously and can find better minima since optimization is informed by the end-task.

Notably, Gururangan et al. (2020) have demonstrated the benefits of continued pre-training of generalist models using data that is similar to that of the end-task. Their approach is formalized into two classes: Domain Adaptive Pre-training (DAPT) and Task Adaptive Pretraining (TAPT) where further stages of pre-training of generalist models are conducted on domain- and task-specific data,

respectively. DAPT and TAPT exploit the fact that *we often know the end-task beforehand*, and so we can make specific choices about our pre-training regimen to improve end-task performance.

However, in both pre-training for generalist models and continued pre-training, the training procedure itself does not explicitly incorporate the *end-task objective function*. Because of this, practitioners have to be careful with their choice of auxiliary tasks, the order in which they are trained on, and the early-stopping criteria for each pre-training stage so as to actually achieve good downstream end-task performance (Gururangan et al., 2020; Dery et al., 2021). In the absence of principled criteria to make these difficult design choices, it is common to instead resort to the computationally demanding heuristic of pre-training on as much data as possible for as long as possible.

In this paper, we raise the following question: "*In settings where we have a particular end-task in mind, should we be **pre-**training at all?*". We define **pre-**training as any form of task-agnostic training that a model undergoes before it is finally fine-tuned on the end-task of interest. As a first milestone in addressing the larger question posed above, we explore the ubiquitous continued pre-training setting (Gururangan et al., 2020; Aghajanyan et al., 2021). Specifically, our paper questions the wisdom of having disjoint *further pre*-training then fine-tuning steps on a *generalist* model. In response, we advocate for an alternative approach in which we directly introduce the end-task objective of interest into the learning process. This results in a suite of end-task aware methods called TARTAN (end-**T**ask **A**wa**R**e **Tr**Aini**N**g). Our formulations incorporate both unsupervised auxiliary objectives traditionally used in NLP pre-training (such as masked language modeling as in Devlin et al. (2018)) *and* the end-task objective, followed by an optional fine-tuning step on the end-task. We motivate TARTAN experimentally in the continued pre-training setting and based on this, we make the following contributions to the literature on leveraging auxiliary tasks and data:

- In lieu of standard end-task agnostic continued pre-training, we suggest introducing the end-task objective into the training process via multi-task learning (Caruana, 1997; Ruder, 2017). We call this procedure **M**ulti-**T**asking end-**T**ask **A**wa**R**e **Tr**Aini**N**g (MT-TARTAN) (Section 3.1). MT-TARTAN is a simple yet surprisingly effective alternative to task-agnostic pre-training. In Section 5, we demonstrate that MT-TARTAN significantly improves performance and data efficiency over Gururangan et al. (2020)'s results. It also obviates the need for fickle hyper-parameter tuning through direct optimization of validation performance.
- To allow more fine-grained control of the end-task over the auxiliary tasks, in Section 3.2, we present an online meta-learning algorithm that learns adaptive multi-task weights with the aim of improving final end-task performance. Our **META**-learning end-**T**ask **A**wa**R**e **Tr**Aini**N**g (META-TARTAN) allows us to robustly modulate between multiple objectives and further improves performance over MT-TARTAN .
- A naive implementation of META-TARTAN based on first-order meta-learning analysis results in a sub-optimal algorithm that ignores all tasks except the end-task. We trace this problem to the use of a single model training head for computing both the end-task training loss and meta-objective (end-task validation loss). To guard against this pathological solution, we introduce a separate model head for computing the meta-objective. In Section 3.3, we justify this simple-to-implement fix and validate its practical efficacy in Section 5.

Our results suggest that TARTAN may be an attractive alternative to the continued pre-training paradigm, and further research into the place of *pre*-training in end-task aware settings is warranted.

## 2 FORMALIZING PRE-TRAINING AND CONTINUED PRE-TRAINING

Consider a dataset $D = \{(x_i, y_i)_{i \in [m]}\}$ consisting of $m$ labelled examples. We define a task as an objective function and dataset pair: $T = \{\mathcal{L}(\cdot), D\}$. $M_\theta$ is a model parameterized by $\theta$. The objective function $\mathcal{L}(y_i, M_\theta(x_i))$ evaluates how well a model prediction $M_\theta(x_i)$ fits the true label $y_i$, such as cross-entropy loss in the case of classification. Note that the task dataset, $D$, is typically decomposed into the sets $(D^{\text{train}}, D^{\text{val}}, D^{\text{test}})$. $D^{\text{train}}$ is the set of examples used for model training whilst $D^{\text{test}}$ is used for final task evaluation. The validation set, $D^{\text{val}}$, is typically used for model selection but it is also frequently used in meta-learning to define the meta-objective – $\mathcal{L}^{val}$.

Given a specific end-task $T^*$, our aim is to improve performance on $T^*$ (as measured by the model loss on $D^{\text{test}}_{T^*}$) by leveraging auxiliary tasks $\mathbb{T}_{\text{aux}} = \{T_1, \ldots, T_n\}$. Note that we do not particularly

care about the performance of any of the tasks in $\mathbb{T}_{\text{aux}}$. We are willing to sacrifice performance on $\mathbb{T}_{\text{aux}}$ if it improves performance on $T^*$.

From the perspective of model architecture, there are several ways to leverage $\mathbb{T}_{\text{aux}}$. We focus on the simple but widely-used parameter sharing setting. Here, all tasks share a model body $\theta_{\text{body}}$ but each task $T_i$ has its own head $\phi^i$ for prediction. We denote the head belonging to $T^*$ as $\phi'$. Thus $\theta = \left[\theta_{\text{body}}; \left(\phi^1, \ldots, \phi^n, \phi'\right)\right]$ and $\theta_{\text{body}}$ is reusable across new tasks.

## 2.1 PRE-TRAINING

Pre-training is when a model is first trained on $\mathbb{T}_{\text{aux}}$ before performing a final fine-tuning phase on $T^*$. The motivation behind pre-training is that learning $\mathbb{T}_{\text{aux}}$ first hopefully captures relevant information that can be utilized during training of $T^*$. This desire has led to the proliferation of generalist pre-trained models like BERT (Devlin et al., 2018), RoBERTa (Liu et al., 2019) and GPT-3 (Brown et al., 2020) that have been trained on copious amounts of data. Generalist models have been widely successful at improving downstream task performance when used as initilization.

We can formalize the pre-training procedure as follows:

$$\theta_0 = \text{argmin}_\theta \left( \sum_{T_i \in \mathbb{T}_{\text{aux}}} \mathcal{L}_{T_i}(\theta) \right) \tag{1}$$

In Equation 1, we seek a point $\theta_0$ that achieves minimal loss on the tasks in $\mathbb{T}_{\text{aux}}$. We hope that $\theta_0$ will be a good starting point for gradient descent on $T^*$. Let $g(\theta_0)$ represent the set of end-points of stochastic gradient descent on an initialization, $\theta_0$. Stochastic gradient descent from the same initialization can produce different end-points due to differences in hyper-parameters like learning rate, batch size and order, as well as regularization strength. We can write the fine-tuning phase as:

$$\theta^* = \text{argmin}_{\{\theta \, \in \, g(\theta_0)\}} \mathcal{L}_{T^*}(\theta) \tag{2}$$

Note that pre-training is *end-task agnostic*: the pre-training Equation 1 occurs entirely before training on the end-task Equation 2, and does not explicitly incorporate the end-task objective, $T^*$. Since there is no awareness of the end-task during pre-training it is important to carefully choose $\mathbb{T}_{\text{aux}}$ so that pre-training actually results in improved performance on $T^*$ (Wang et al., 2018a). For text data, past work has found left-to-right language modeling (Peters et al., 2017) and masked language modeling (MLM) (Devlin et al., 2018) to be good choices to include in $\mathbb{T}_{\text{aux}}$.

## 2.2 CONTINUED PRE-TRAINING

Recent work (Beltagy et al., 2019; Gururangan et al., 2020; Lee et al., 2020) showed that downstream performance on $T^*$ can be improved by further adapting generalist models via continued pre-training on a more relevant set of auxiliary tasks. This is equivalent to sequentially performing multiple steps of Equation 1, with different $\mathbb{T}_{\text{aux}}$, before finally performing Equation 2 on $T^*$.

**Domain and Task Adaptive Pre-training** Gururangan et al. (2020) present Domain Adaptive Pre-Training (DAPT) and Task Adaptive Pre-Training (TAPT) as methods for continued pre-training. During DAPT, a generalist model is further pre-trained on an unsupervised objective with large amounts of data from the same domain as the end-task. TAPT also pre-trains with the same unsupervised objective as DAPT, but on the actual dataset of the end-task. Gururangan et al. (2020) find that performance can be further improved by chaining objectives, DAPT first, followed by TAPT.

Though TAPT and DAPT do not directly incorporate the end-task objective during training, it still indirectly informs both the choice of pre-training data and the order in which the pre-training tasks are trained on. Below, we explore stronger versions of this influence.

## 3 END-TASK AWARE TRAINING (TARTAN)

In this section, we argue for the end-task to be added directly into the training process to create explicit interactions between $T^*$ and $\mathbb{T}_{\text{aux}}$.

### 3.1 END-TASK AWARE TRAINING VIA MULTI-TASKING (MT-TARTAN)

We propose to directly incorporate knowledge of the end-task by multi-tasking $T^*$ together with $\mathbb{T}_{\text{aux}}$, before optionally fine-tuning on $T^*$ exclusively. To this end, we introduce a set of task weights $\mathbf{w} = (w^*, w_1, \cdots, w_{|\mathbb{T}_{\text{aux}}|})$ satisfying $w^* + \sum_i w_i = 1$, to modulate between the different losses. Our new formulation is:

$$\theta_0 = \text{argmin}_\theta \ \mathcal{L}_{\text{total}}(\theta, \mathbf{w}) = \text{argmin}_\theta \left( w^* \mathcal{L}_{T^*}(\theta) + \sum_i w_i \mathcal{L}_{T_i}(\theta) \right) \tag{3}$$

Here, Equation 3 replaces Equation 1 and can be followed by the optional fine-tuning stage of Equation 2. Note that this formulation fixes the tasks weights $\mathbf{w}$ throughout the training process. We call this formulation End-task Aware Training via Multi-tasking (MT-TARTAN) because we introduce the end-task **directly** into the training procedure, and do so by **multi-tasking** it with $\mathbb{T}_{\text{aux}}$.

MT-TARTAN allows us to prioritize performance on $T^*$ in several ways. First, we can weight the end-task higher than all the other auxiliary tasks. Also, during training, we can monitor $\mathcal{L}_{T^*}$ on the end-task validation set and early stop when it plateaus; even if the auxiliary tasks have not yet converged. This is not possible during standard pre-training because we do not train $T^*$ and so it performs at random before we actually start fine-tuning. Early stopping on $T^*$ can represent significant computational savings over end-task agnostic pre-training when the savings in data-efficiency supercede the extra overhead of end-task aware gradient descent steps.

### 3.2 END-TASK AWARE TRAINING VIA META-LEARNING (META-TARTAN)

MT-TARTAN, DAPT and TAPT, all share the same drawback: they implicitly assume that the auxiliary tasks have static importance to the end-task over the lifetime of its training, either by being end-task agnostic (DAPT and TAPT) or by having static task weights (MT-TARTAN). With MT-TARTAN, an additional drawback noted by Wang et al. (2019); Yu et al. (2020) is that multi-tasking can negatively impact task performance compared to isolated training. These shortcomings motivate the formulation of an adaptive algorithm that can mitigate the negative influence of some tasks whilst responding to the changing relevance of auxiliary tasks over the lifetime of end-task training.

As they stand, the pre-training equation pair (Equations 1, 2) and the MT-TARTAN pair (Equations 2, 3) are decoupled. The inner-level variables of the pre-training phase do not depend on the outer-level variables of the fine-tuning phase. Thus the equation pairs are typically solved sequentially. We propose to tightly couple Equations 2 and 3 by formulating jointly learning $\mathbf{w}$ and $\theta_0$ as a bi-level optimization problem. A bi-level formulation allows us to leverage meta-learning (Schmidhuber, 1995) techniques to learn adaptive task weights which capture variable auxiliary task importance whilst mitigating the contribution of harmful tasks. We propose a meta-learning algorithm in the mold of Model Agnostic Meta-Learning (MAML) (Finn et al., 2017) to learn task weights. As a bi-level problem, this can be formulated as :

$$\theta^*, \mathbf{w}^* = \text{argmin}_{\{\theta \,\in\, g(\theta_0), \, \mathbf{w}\}} \mathcal{L}_{T^*}(\theta) \tag{4}$$

where

$$\theta_0 = \text{argmin}_\theta \ \mathcal{L}_{\text{total}}(\theta, \mathbf{w}) = \text{argmin}_\theta \left( w^* \mathcal{L}_{T^*}(\theta) \ + \sum_{T_i \in \mathbb{T}_{\text{aux}}} w_i \mathcal{L}_{T_i}(\theta) \right) \tag{5}$$

We want to jointly learn $\mathbf{w}$, with $\theta_0$, such that taking a gradient descent step modulated by $\mathbf{w}$ leads to improvement in end-task generalization. We use performance on the end-task validation set $(D_{T^*}^{\text{val}})$ as a meta-objective to train $\mathbf{w}$. Performance on $D_{T^*}^{\text{val}}$ serves as a stand-in for end-task generalization performance whilst also naturally capturing the asymmetrical importance of $T^*$.

Our joint descent algorithm proceeds as follows. At each timestep $t$, we hold the task weights fixed and update $\theta_t$ based on $\nabla_\theta \mathcal{L}_{\text{total}}(\theta_t, \mathbf{w})$. We then proceed to update $\mathbf{w}$ via gradient descent on the end-task validation loss at $\theta_{t+1}$. For this, we derive an approximation for $\nabla_\mathbf{w} \mathcal{L}_{T^*}^{val}(\theta_{t+1}, \mathbf{w})$ below:

$$\mathcal{L}_{T^*}^{val}(\theta_{t+1}(\mathbf{w})) = \mathcal{L}_{T^*}^{val} \left( \theta_t - \beta \left( w^* \nabla \mathcal{L}_{T^*} + \sum_i w_i \nabla \mathcal{L}_{T_i} \right) \right)$$

$$\approx \mathcal{L}_{T^*}^{val}(\theta_t) - \beta \left( w^* \nabla \mathcal{L}_{T^*} + \sum_i w_i \nabla \mathcal{L}_{T_i} \right)^T \nabla \mathcal{L}_{T^*}^{val}(\theta_t)$$

We can take the gradient of the above first-order approximation w.r.t an individual weight $w_i$. This tells us how to update $w_i$ to improve the meta-objective.

$$\frac{\partial \mathcal{L}_{T^*}^{val}(\theta_{t+1}(\mathbf{w}))}{\partial w_i} \approx -\beta\big(\nabla \mathcal{L}_{T_i}\big)^T \big(\nabla \mathcal{L}_{T^*}^{val}(\theta_t)\big) = -\beta\big(\nabla \mathcal{L}_{T_i}\big)^T \big(\nabla \mathcal{L}_{T^*}^{val}([\theta_{\text{body}}, \phi']_t)\big) \quad (6)$$

In Equation 6, we explicitly specify $[\theta_{\text{body}}, \phi']_t$ because computing losses on $T^*$ depend on only these parameters. $\mathcal{L}_{T_i}$ depends solely on $[\theta_{\text{body}}, \phi^i]_t$ but we leave this out to avoid notation clutter.

Our analysis above is similar to that of Lin et al. (2019) with one key difference: we learn a weighting for the main task $w^*$ too. This ability to directly modulate $T^*$ allows us to capture the fact that at certain stages in training, auxiliary tasks may have greater impact on end-task generalization than the end-task's own training data. This choice also allows us to control for over-fitting and the influence of bad (mislabelled or noisy) training data.

### 3.3 INTRODUCING A SEPARATE CLASSIFICATION HEAD FOR META-LEARNING

Observe that from Equation 6, updates for $w \neq w^*$ involve gradients computed from different model heads $\phi^i$ and $\phi'$ whilst for $w^*$, we are taking the dot product of gradients from the same end-task head $\phi'$. As we will show empirically in Section 5.4, computing weight updates this way creates a strong bias towards the primary task, causing $w^*$ to rail towards 1 whilst the other weights dampen to 0, which may be sub-optimal in the long run.

Intuitively, this short-horizon (greedy) (Wu et al., 2018) behavior makes sense: the quickest way to make short-term progress (improve $\mathcal{L}_{T^*}^{val}(\theta_{t+1})$) is to descend solely on $T^*$. More formally, the greedy approach arises because we derive $\nabla_{w_i} \mathcal{L}_{T^*}^{val}(\theta_{t+1})$ in Equation 6 as a proxy for the gradient at $\theta^*$, the outer-loop end-point in Equation 4. Variations of this substitution are common in the meta-learning literature (Finn et al., 2017; Liu et al., 2018; Nichol et al., 2018) because it is computationally infeasible to train a model to convergence every time we wish to compute $\nabla_{w_i} \mathcal{L}_{T^*}^{val}(\theta^*)$.

To remedy the greedy solution, instead of estimating $\nabla_\theta \mathcal{L}_{T^*}$ and $\nabla_\theta \mathcal{L}_{T^*}^{val}$ from the same classification head (Equation 6), we introduce a special head $\phi^*$ for computing the meta-objective. Specifically, instead of trying to compute $\theta^*$, we approximate it by fixing the body of the network $\theta_{\text{body}}$ and training the randomly initialized head $\phi^*$ to convergence on a subset of the end-task training data. We do this every time we wish to estimate $\nabla_{w_i} \mathcal{L}_{T^*}^{val}(\theta^*)$. Introducing $\phi^*$ eliminates the strong positive bias on $w^*$ and enables us to compute a better proxy for the meta-gradient at $\theta^*$:

$$\frac{\partial \mathcal{L}_{T^*}^{val}(\theta^*(\mathbf{w}))}{\partial w_i} \approx \big(\nabla_\theta \mathcal{L}_{T_i}\big)^T \big(\nabla_\theta \mathcal{L}_{T^*}^{val}([\theta_{\text{body}}; \phi^*]_t)\big) \quad (7)$$

Equation 7 represents a simple-to-implement alternative to Equation 6. We provide a more detailed justification for Equation 7 in Appendix A.1. In Section 5.4, we empirically validate that the transition from Equation 6 to 7 improves performance whilst mitigating pathological solutions. Our approach of creating $\phi^*$ for approximating the meta-objective (down-stream validation performance) is inspired by Metz et al. (2018), who use a similar technique to construct a meta-objective for evaluating the quality of unsupervised representations.

**Please see Algorithm 1 in Appendix A.3 for details about META-TARTAN.**

## 4 EXPERIMENTAL SETUP

**Setting**[1]   Though our algorithms and methodology can be directly applied to both continued pre-training (Section 2.2) and pre-training from scratch (Section 2.1) of *generalist* models, we focus on the former scenario. This is because the continued pre-training setting is more common amongst everyday practitioners as it is less computationally demanding. It thus lends itself more easily to exploration under a realistic computational budget. In Appendix A.4, we show that end-task aware training from scratch is viable by studying a simple computer vision setting. Concurrent work by Yao et al. (2021) shows that from-scratch end-task aware training for NLP problems is viable.

---

[1]Code will be released at https://github.com/ldery/TARTAN

In keeping with previous work (Devlin et al., 2018; Gururangan et al., 2020), we focus on $\mathbb{T}_{aux}$ as a set of MLM tasks on varied datasets. In the case of DAPT and our end-task aware variants of it, $\mathbb{T}_{aux}$ is an MLM task with data from the domain of the end-task. For TAPT, $\mathbb{T}_{aux}$ is an MLM task with data from the end-task itself. DAPT, TAPT and DAPT+TAPT (chained pre-training with DAPT followed by TAPT) will serve as our baseline continued pre-training approaches. We will compare these baselines to their end-task aware variants that use MT-TARTAN and META-TARTAN.

**Datasets** Our experiments focus on two domains: computer science (CS) papers and biomedical (BIOMED) papers. We follow Gururangan et al. (2020) and build our CS and BIOMED domain data from the S2ORC dataset (Lo et al., 2019). We extract 1.49M full text articles to construct our CS corpus and 2.71M for our BIOMED corpus. Under both domains, our end-tasks are *low-resource* classification tasks. Using low-resource tasks allows us to explore a setting where pre-training can have a significant impact. Under the CS domain, we consider two tasks: ACL-ARC (Jurgens et al., 2018) and SCIERC (Luan et al., 2018). ACL-ARC is a 6-way citation intent classification task with 1688 labelled training examples. For SCIERC, the task is to classify the relations between entities in scientific articles. This task has 3219 labelled examples as training data. We choose CHEMPROT (Kringelum et al., 2016) as the classification task from the BIOMED domain. This task has 4169 labelled training examples and the goal is to classify chemical-protein interactions. More details of these datasets can be found in Table 2 of Gururangan et al. (2020). Gururangan et al. (2020) evaluate against all 3 tasks and their available code served as a basis on which we built MT-TARTAN and META-TARTAN.

**Model Details** We use a pre-trained RoBERTa$_{base}$ (Liu et al., 2019) as the shared model base and implement each task as a separate multi-layer perceptron (MLP) head on top of this pre-trained base. As in Devlin et al. (2018), we pass the `[CLS]` token embedding from RoBERTa$_{base}$ to the MLP for classification.

**Training Details** For DAPT and TAPT, we download the available pre-trained model bases provided by Gururangan et al. (2020). To train thier corresponding classification heads, we follow the experimental setup described in Appendix B of Gururangan et al. (2020).

Performing end-task aware training introduces a few extra hyper-parameters. We fix the other hyper-parameters to those used in Gururangan et al. (2020). MT-TARTAN and META-TARTAN introduce joint training of a classification head for the end-task $T^*$. We experiment with batch sizes of 128, 256 and 512 for training this head. We try out learning rates in the set $\{10^{-3}, 10^{-4}, 10^{-5}\}$ and drop out rates of $\{0.1, 0.3\}$. For META-TARTAN since we are now learning the task weights, $\mathbf{w}$, we test out task weight learning rates in $\{10^{-1}, 5 \times 10^{-2}, 3 \times 10^{-2}, 10^{-2}\}$. Note that for all MT-TARTAN experiments we use equalized task weights $\frac{1}{|\mathbb{T}_{aux}|+1}$. A small grid-search over a handful of weight configurations did not yield significant improvement over the uniform task weighting. We use the Adam optimizer (Kingma & Ba, 2014) for all experiments.

As mentioned in section 3.3, we train as separate meta-classification head, $\phi^*$, to estimate the validation meta-gradients. To estimate $\phi^*$, we use batch sizes of $\{16, 32\}$ samples from $T^*$'s train set. We regularize the meta-head with $l_2$ weight decay and set the decay constant to 0.1. We use a learning rate $10^{-3}$ to learn the meta-head. We stop training $\phi^*$ after 10 gradient descent steps.

## 5 RESULTS AND DISCUSSION

In this section, we will discuss the results of comparing our models against DAPT and TAPT baselines.[2] Broadly, we demonstrate the effectiveness of end-task awareness as improving both performance and data-efficiency.

---

[2]Our results are slightly different from those presented in Table 5 of Gururangan et al. (2020) in terms of absolute values but the trends observed there still hold here. We attribute these differences to (1) minor implementation differences, and (2) averaging performance over ten seeds instead of five as used in the original paper in order to more strongly establish statistical significance. We observe slightly lower performance on ACL-ARC and SCIERC tasks due to these changes and higher performance on CHEMPROT.

| Domain | Task | RoBERTa | TAPT | MT-TARTAN | META-TARTAN |
|--------|------|---------|------|-----------|-------------|
| CS | ACL-ARC | $66.03_{3.55}$ | $67.74_{3.68}$ | $\mathbf{70.48_{4.42}}$ | $70.08_{4.70}$ |
|  | SCIERC | $77.96_{2.96}$ | $79.53_{1.93}$ | $\mathbf{80.81_{0.74}}$ | $\mathbf{81.48_{0.82}}$ |
| BIOMED | CHEMPROT | $82.10_{0.98}$ | $82.17_{0.65}$ | $\mathbf{84.29_{0.63}}$ | $\mathbf{84.49_{0.50}}$ |

Table 1: Comparison of our end-task aware approaches to RoBERTa and TAPT. All end-task aware approaches use TAPT as the auxiliary task. Reported results are test macro-F1, except for CHEMPROT, for which we report micro-F1, following Beltagy et al. (2019). We average across 10 random seeds, with standard deviations as subscripts. Statistically significant performance (p-value from permutation test $< 0.05$), is boldfaced. See A.2, A.5 for more details about this table

## 5.1 END-TASK AWARENESS IMPROVES OVER TASK-AGNOSTIC PRE-TRAINING

Table 1 compares TAPT to its end-task aware variants. As in Gururangan et al. (2020), we observe that performing task adaptive pre-training improves upon just fine-tuning RoBERTa. However, note that introducing the end-task by multi-tasking with the TAPT MLM objective leads to a significant improvement in performance. This improvement is consistent across the 3 tasks we evaluate against. We find that both MT-TARTAN and META-TARTAN achieve similar results in this setting.

## 5.2 END-TASK AWARENESS IMPROVES DATA-EFFICIENCY

Gururangan et al. (2020) train DAPT on large amounts of in-domain data to achieve results competitive with TAPT. They use 7.55 billion tokens for the BIOMED domain and 8.10 billion for the CS domain. This is on average over $10^4\times$ the size of the training data of our end-tasks of interest. The large amount of data required to train a competitive DAPT model represents a significant computational burden to the every-day practitioner. This begets the question: *are such large amounts of auxiliary data necessary for achieving good downstream performance?* To answer this, we train DAPT and its TARTAN version on variable amounts of data for both SCIERC and ACL-ARC tasks.

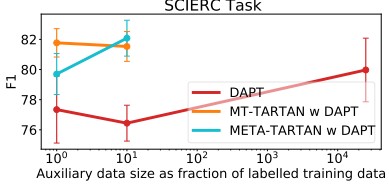 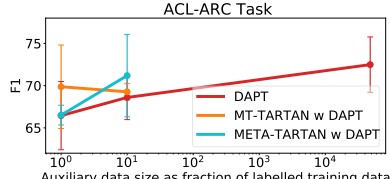

Figure 2: Compared to DAPT, TARTAN makes more efficient use of data. Large standard deviations are a result of the heterogeneity of the domain data used and the fact that our tasks are low-resource.

**TARTAN is more data-efficient than DAPT** In Figure 2, we focus on training on a small fraction of available domain data $n = \{10^0, 10^1\} \times |\text{Train}|$ for the DAPT auxiliary task. Full domain data is $n' \approx 10^4 \times |\text{Train}|$. This relatively low auxiliary data regime represents a realistic setting that is akin to those encountered by everyday practitioners who are likely to be computationally constrained. As can be seen in Figure 2, on the ACL-ARC task, META-TARTAN matches the performance of DAPT when the sizes of the domain data and end-task data are of the same order ($10^0$). At this data size, META-TARTAN supersedes DAPT on the SCIERC task. When trained on $10\times$ more auxiliary data, META-TARTAN supersedes DAPT in performance on both tasks. On the ACL-ARC task, META-TARTAN achieves $71.19_{4.88}$, which is close to DAPT's performance of $72.49_{3.28}$ using more than $10^3\times$ auxiliary data. These results indicate that end-task awareness can improve data-efficiency and in this case, improvements are on the order of $1000\times$.

| Domain | Task | DAPT | DAPT+TAPT | MT-TARTAN | META-TARTAN |
|--------|------|------|-----------|-----------|-------------|
| CS | ACL-ARC | $68.60_{2.62}$ | $69.12_{5.76}$ | $71.58_{1.65}$ | $71.05_{2.37}$ |
|  | SCIERC | $76.44_{1.19}$ | $77.62_{1.38}$ | $\mathbf{81.02_{1.24}}$ | $\mathbf{81.41_{1.70}}$ |
| BIOMED | CHEMPROT | $80.76_{0.54}$ | $78.22_{0.74}$ | $\mathbf{83.77_{0.60}}$ | $\mathbf{83.38_{0.89}}$ |

Table 2: We use $n = 10 \times |\text{Train}|$, a small fraction the full domain data which is $> 10^4 \times |\text{Train}|$. TARTAN methods are trained on both DAPT and TAPT. We average performance across 10 seeds. Statistically significant performance is boldfaced. See A.2, A.6 for more details about this table.

**TARTAN is more data-efficient than DAPT+TAPT** Table 2 compares DAPT and DAPT+TAPT (DAPT followed by TAPT) to *-TARTAN which multi-task DAPT, TAPT and the end-task. MT-TARTAN and META-TARTAN significantly outperform DAPT and DAPT+TAPT in 2 of the tasks whilst giving higher average performance in the ACL-ARC task. We thus conclude that **end-task awareness allows us to get a greater performance boost out of the same amount of data**.

We explore the data efficiency of TARTAN methods even further by comparing the relatively data-poor versions of MT-TARTAN and META-TARTAN above ($n = 10 \times$ |Train|) to the DAPT and DAPT+TAPT variants trained on all the available domain data ($n' \approx 10^4 \times$ |Train|). We can see from Table 3 that for the CS domain, our end-task aware variants come close to (ACL-ARC) and even supersede (SCIERC) the end-task agnostic variants though trained with $\approx 1000\times$ less data. For BIOMED domain (CHEMPROT task), increasing the amount of data drastically improves the performance of end-task agnostic variants compared to MT-TARTAN and META-TARTAN trained on much less data.

Zhang et al. (2020) show that different tasks exhibit sigmoid-like curves in terms of how much pretraining data is required to achieve good results before performance levels off. We contextualize Tables 2 and 3 within said work and posit that the CHEMPROT task intrinsically requires much more data (compared to our other tasks) before performance begins to improve appreciably.

| Task | $\mathbf{DAPT}_{full}$ | $\mathbf{+TAPT}_{full}$ |
|---|---|---|
| ACL-ARC | $72.49_{3.28}$ | $73.79_{1.75}$ |
| SCIERC | $79.97_{2.11}$ | $80.00_{1.08}$ |
| CHEMPROT | $86.54_{1.05}$ | $87.24_{0.81}$ |

Table 3: DAPT and DAPT+TAPT runs on all domain data available.

## 5.3 META-TARTAN MORE EFFECTIVELY UTILIZES OUT-OF-DISTRIBUTION AUXILIARY DATA OVER MT-TARTAN

| TARTAN | ACL-ARC | SCIERC | CHEMPROT |
|---|---|---|---|
| MT | $69.27_{0.96}$ | $81.53_{0.99}$ | $80.26_{3.79}$ |
| META | $\mathbf{71.19}_{4.88}$ | $\mathbf{82.08}_{1.19}$ | $\mathbf{82.31}_{0.75}$ |

Table 4: All methods use only DAPT as auxiliary task. We use $n = 10 \times$ |Train|. We report averages across 3 random seeds. Best average task performance is bolded.

We have seen that leveraging TAPT (Table 1 and 2) leads MT-TARTAN and META-TARTAN to perform similarly. The advantage of learning adaptive weights becomes pronounced in the DAPT only setting. Whilst TAPT uses the end-task's own training data for masked language modelling, DAPT uses heterogeneous domain data whose impact on the end-task performance is less clear. Notice from Table 4 that when required to rely solely on domain data for auxiliary tasking, META-TARTAN improves performance over MT-TARTAN. We attribute META-TARTAN's improvement over MT-TARTAN to its ability to more flexibly adapt to incoming data of variable utility to the end-task.

## 5.4 TASK WEIGHTING STRATEGIES DISCOVERED BY META-LEARNING

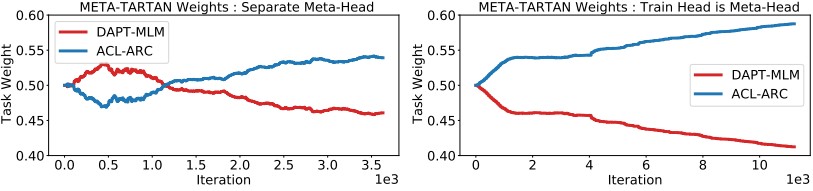

Figure 3: Having a separate classification head for computing meta-gradients is important. Using the same head as when training up-weights the end-task and under-utilizes auxiliary tasks.

To illustrate the importance of the separate classification head $\phi^*$ for computing the meta-signal for the task weights (described in Section 3.3), we run META-TARTAN experiments with ACL-ARC as the end-task and DAPT as the auxiliary task. We compare using either a separate ($\phi^*$) or the same ($\phi'$) classification head for calculating the meta-gradient. Figure 3 plots the task weightings learned in each setting during training. We can clearly see that using a separate head counteracts the pathological solution of down-weighting all tasks that are not $T^*$ and as a result, improves performance: **a delta of 1.7 F1 points in this case**. The strategy discovered by META-TARTAN presents an interesting contrast to classical pre-training: whilst the initial phase of classical pre-training involves

Figure 4: The meta-learned task weightings show similar trajectories across different end-tasks.

solely the auxiliary task, early in training, META-TARTAN up-weights the auxiliary task but does not fully zero out the end-task. Later in training, we see leveling off of weights instead of railing the end-task to 1 as in classical pre-training.

Next, we plot a similar graph for using both DAPT and TAPT across our three tasks in Figure 4. From the figure, it is apparent that META-TARTAN discovers similar task-weighting strategies across different end-tasks. This suggests that the MLM objective and META-TARTAN's strategy for learning task weights are generic enough to induce similar behaviours across tasks. In general, DAPT is significantly up-weighted compared to the end-task and TAPT. Note that the TAPT + ACL-ARC task weights (Figure 4) has the same approximate trajectory as ACL-ARC task weight in Figure 3. It seems important to assign high weight to the task data (Figure 3) but not necessarily all of it needs to go to the actual task loss (Figure 4). We hypothesize that the diversity in the domain data counteracts overfitting to the end-task data and results in DAPT being up-weighted.

## 6 RELATED WORK

Multi-task learning can be traced back to seminal work by Caruana (1995), Caruana (1997), and has since been the subject of a flourishing literature, recent surveys of which can be found in Ruder (2017) or Zhang & Yang (2021). In NLP, while initial work from Collobert & Weston (2008) already showed the benefits of multi-task learning, it has only recently become a central topic in the field, with the advent of multi-task benchmarks (Wang et al., 2018b; McCann et al., 2018).

Pre-training is where a machine learning model is first trained on a generic, data-rich task before being fine-tuned on an end-task. In NLP this practice dates back to the use of pre-trained word embeddings (Turian et al., 2010; Mikolov et al., 2013) and later pre-trained encoders (Kiros et al., 2015; Dai & Le, 2015). Peters et al. (2018) and Howard & Ruder (2018) heralded a renaissance of pre-training before BERT (Devlin et al., 2018) and its many offshoots (Liu et al., 2019; Yang et al., 2019; Lewis et al., 2019) cemented it as the de facto standard for modern NLP.

Meta-learning dates back to early work from Schmidhuber (1995); Thrun (1998). More relevant to our work is gradient-based meta-learning for solving bi-level optimization problems, first popularized by Finn et al. (2017) and followup work (Nichol et al., 2018; Rajeswaran et al., 2019) for few-shot learning. This method has transferred to a variety of applications such as architecture search (Liu et al., 2018) and model poisoning (Kurita et al., 2020).

## 7 CONCLUSION

We have advocated for a paradigm shift in the way we approach pre-training. We have motivated making pre-training more end-task aware when the end-task is known in advance. Our work introduced two novel end-task aware training algorithms: End-task Aware Training via Multi-tasking (MT-TARTAN) and End-task Aware Training via Meta-learning (META-TARTAN). In Section 5, we demonstrated the ability of our proposed algorithms to improve performance and data-efficiency over their end-task agnostic counterparts.

This work suggests several promising directions for future work. Instead of learning coarse task level weights, can further performance improvements be achieved via finer-grained example level weighting as in Wang et al. (2020)? Can meta-learning algorithms like META-TARTAN enable more effective utilization of previously discarded (Aroca-Ouellette & Rudzicz, 2020) pre-training auxiliary tasks like Next Sentence Prediction (NSP) (Devlin et al., 2018)? We hope this work spurs conversation around these questions and many more.

## 8 ACKNOWLEDGEMENTS

This work was supported in part by DSO National Laboratories, an ENS-CFM Data Science Chair, DARPA FA875017C0141, the National Science Foundation grants IIS1705121, IIS1838017, IIS2046613 and IIS-2112471, an Amazon Web Services Award, a Facebook Faculty Research Award, funding from Booz Allen Hamilton Inc., and a Block Center Grant. Any opinions, findings and conclusions or recommendations expressed in this material are those of the author(s) and do not necessarily reflect the views of any of these funding agencies.

## 9 ETHICS STATEMENT

Our work introduces new algorithms but leverages pre-existing datasets and models. Overall, this work inherits some of the risk of original work upon which it is implemented. Algorithms for continued training such as TAPT and DAPT necessitate per-task training of unsupervised objectives which result in corresponding green-house emissions due to energy consumption (Strubell et al., 2019). However, as shown in Sections 3 and 5, our new compute-efficient algorithms greatly increase the data efficiency of these algorithms, reducing these harms as well as the various harms associated with labor for data-collection (Jo & Gebru, 2020). Also, since our work is set in the context of pre-existing datasets and models (Section 4), we recognize that any ethical issues that have been revealed in these (such as bias (Bender et al., 2021) or privacy leakage (Carlini et al., 2021)) may also propagate to models trained using our work, and mitigation strategies such as Schick et al. (2021); Liang et al. (2021) may be necessary. Finally, there is a potential risk in META-TARTAN that leveraging a validation set for defining the meta-objective could amplifying bias that exists in this data split, although this is done indirectly through task weighting and hence we believe that this risk is small.

## 10 REPRODUCIBILITY STATEMENT

We pledge to release the source-code for this project to improve the ease of reproducibility of our results by the NLP and machine learning communities. In Section 4, we have specified details about datasets, training regimes and models to allow anyone who wishes to reproduce our results without our original source code to do so. Our discussion of the algorithmic and evaluation details can be found in Appendices A.1, A.3 and A.2. As we noted in 4, we build off of Gururangan et al. (2020)'s implementations which can be found at https://github.com/allenai/dont-stop-pretraining.

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

# A   APPENDIX

## A.1   JUSTIFYING THE INTRODUCTION OF A META-HEAD

*Proof.* To arrive at Equation 7 we start with the closed form solution for $\nabla_{w_i}\mathcal{L}_{T^*}^{val}(\theta^*)$ and then introduce approximations in order to produce Equation 7. First, note that :

$$\frac{\partial\mathcal{L}_{T^*}^{val}(\theta^*(\mathbf{w}))}{\partial w_i} = \left(\nabla_\theta\mathcal{L}_{T^*}^{val}(\theta^*(\mathbf{w}))\right)^T \left(\nabla_{w_i}\theta^*(\mathbf{w})\right) \quad \texttt{[Chain rule]} \tag{8}$$

To get $\nabla_{w_i}\theta^*(\mathbf{w})$ we invoke the Cauchy Implicit Function Theorem (IFT) as with Lorraine et al. (2020); Navon et al. (2020); Liao et al. (2018):

$$
\begin{aligned}
\nabla_{w_i}\theta^*(\mathbf{w}) &= \left[\nabla_\theta^2\mathcal{L}_{\text{total}}(\theta^*(\mathbf{w}))\right]^{-1} \left[\nabla_{w_i}\nabla_\theta\mathcal{L}_{\text{total}}(\theta^*(\mathbf{w}))\right] \quad \texttt{[IFT]} \\
&= \left[\nabla_\theta^2\mathcal{L}_{\text{total}}(\theta^*(\mathbf{w}))\right]^{-1} \left[\nabla_{w_i}\nabla_\theta\left(w^*\mathcal{L}_{T^*}(\theta^*(\mathbf{w})) + \sum_{T_i\in\mathbb{T}_{\text{aux}}} w_i\mathcal{L}_{T_i}(\theta^*(\mathbf{w}))\right)\right] \\
&= \left[\nabla_\theta^2\mathcal{L}_{\text{total}}(\theta^*(\mathbf{w}))\right]^{-1} \left[\nabla_\theta\mathcal{L}_{T_i}(\theta^*(\mathbf{w}))\right] \quad \texttt{[Only terms with } w_i \texttt{ survive]}
\end{aligned}
$$

Bringing it all together, we get :

$$\frac{\partial\mathcal{L}_{T^*}^{val}(\theta^*(\mathbf{w}))}{\partial w_i} = \left(\nabla_\theta\mathcal{L}_{T^*}^{val}(\theta^*(\mathbf{w}))\right)^T \left(\left[\nabla_\theta^2\mathcal{L}_{\text{total}}(\theta^*(\mathbf{w}))\right]^{-1}\left[\nabla_\theta\mathcal{L}_{T_i}(\theta^*(\mathbf{w}))\right]\right) \tag{9}$$

$\square$

Computing $\nabla_{w_i}\mathcal{L}_{T^*}^{val}(\theta^*)$ from Equation 9 is computationally unwieldy since we would not only have to optimize $\theta$ to convergence for every step of $w_i$ but we would also have to invert the Hessian of a typically large model. Our middle ground between Equations 9 and 6 (Equation 7) makes use of the following approximations:

- We approximate the inverse Hessian with the identity. This approximation is not new; we follow previous work like Lorraine et al. (2020)(Table 3) who explore the use of this approximation because of computational efficiency.

$$\left[\nabla_\theta^2\mathcal{L}_{\text{total}}(\theta^*(\mathbf{w}))\right]^{-1} = \lim_{i\to\infty}\sum_{j=0}^{i}\left(\mathbf{I} - \nabla_\theta^2\mathcal{L}_{\text{total}}(\theta^*(\mathbf{w}))\right)^j \approx \mathbf{I}$$

  We are assuming the contribution of terms with $i > 0$ are negligible.

- Instead of training the whole network to convergence, at each time-step, we fix the body of the network and train a special head $\phi^*$ to convergence on a small batch of end-task training data. We then use $[\theta_{\text{body}};\phi^*]$ as a proxy for $\theta^*$. This is a computationally feasible workaround to training all of $\theta$ to convergence to get a single step gradient estimate. Especially in the continued pre-training setting where a pre-trained *generalist* model like BERT is used as $\theta_{\text{body}}$, this approximation is reasonable. To our knowledge, we are the first to suggest this approximation.

$$\nabla_\theta\mathcal{L}_{T^*}^{val}(\theta^*) \to \nabla_\theta\mathcal{L}_{T^*}^{val}([\theta_{\text{body}};\phi^*])$$

- Above, we have approximated $\theta^* = [\theta_{\text{body}};\phi^*]$. Since $\phi^*$ is only used to evaluate end-task ($T^*$) validation data, it means $\theta$ remains unchanged with respect to the training data for task $T_i$. Thus $\nabla_\theta\mathcal{L}_{T_i}([\theta_{\text{body}};(\phi^*,\dots,\phi^i)]) = \nabla_\theta\mathcal{L}_{T_i}([\theta_{\text{body}};\phi^i]) = \nabla_\theta\mathcal{L}_{T_i}(\theta)$

Bringing it all together, we get Equation 7, repeated here:

$$\frac{\partial\mathcal{L}_{T^*}^{val}(\theta^*(\mathbf{w}))}{\partial w_i} \approx \left(\nabla_\theta\mathcal{L}_{T_i}\right)^T\left(\nabla_\theta\mathcal{L}_{T^*}^{val}([\theta_{\text{body}};\phi^*]_t)\right)$$

## A.2 Calculating p-values from Permutation Test

We used the permutation test (Good, 2005; Dror et al., 2018) to test for statistical significance. For each test, we generate 10000 permutations to calculate significance level. This is sufficient to converge to a stable p-value without being a computational burden. We chose this over the common student t-test because :

1. We have only 10 runs per algorithm and permutation tests are more robust at low sample size

2. Permutation test is assumption free. Student t-tests assume that the samples are normally distributed

3. Permutation test is robust to variance in the samples, so even though error-bars can overlap, we still establish significant differences in the samples. Variance in our results is expected due to small dataset sizes of end-tasks.

## A.3 Algorithm for META-TARTAN

---

**Algorithm 1:** End-task Aware Training via Meta-learning (META-TARTAN)

---

**Require:** $T^*$, $\mathbf{T}_{\text{aux}}$: End-task, Set of auxiliary pre-training tasks
**Require:** $\eta, \beta_1, \beta_2$: Step size hyper-parameters
**Initialize** :
    Pre-trained RoBERTa as shared network body, $\theta_{\text{body}}$
    Task weightings: $w^*, w_i = \frac{1}{|\mathbf{T}_{\text{aux}}|+1}$
**Randomly initialize** :
    end-task head as $\phi'$
    meta head for end-task as $\phi^*$
    task head, $\phi^i$, for each $T_i \in \mathbf{T}_{\text{aux}}$
**while** *not done* **do**

    $B^*_{\text{tr}} \sim T^*_{\text{train}}$ // Sample a batch from end-task

    $g^*_\theta, g^*_\phi \leftarrow [\nabla_\theta, \nabla_{\phi'}]\Big(\mathcal{L}_{T^*}(\theta, \phi', B^*_{\text{tr}})\Big)$ // Get end-task grads

    $g^i_\theta, g^i_\phi \leftarrow [\nabla_\theta, \nabla_{\phi^i}]\Big(\mathcal{L}_{T_i}(\theta, \phi^i, B_i)\Big)$ // Get task grads. $\forall i \in [n]$, $B_i \sim T_i$

    // Learn a new meta head
    $\phi^* \leftarrow$ estimate_meta_head$(B^*_{\text{tr}}, \beta_2, \theta, \phi^*)$ // $B^*_{\text{tr}} \sim T^*_{\text{train}}$
    $g^*_{meta} \leftarrow \nabla_\theta \mathcal{L}_{T^*}(\theta, \phi^*, B^*_{\text{val}})$ // $B^*_{\text{val}} \sim T^*_{\text{val}}$
    // Update task weightings
    $w^* \leftarrow w^* + \eta \cos(g^*_{meta}, g^*_\theta)$
    $w_i \leftarrow w_i + \eta \cos(g^*_{meta}, g^i_\theta)$
    // Update task parameters
    $\alpha^*, \alpha_1, \ldots, \alpha_{|\mathbf{T}_{\text{aux}}|} = \text{softmax}(w^*, w_1, \ldots, w_{|\mathbf{T}_{\text{aux}}|})$
    Update $\theta_{\text{body}} \leftarrow \theta_{\text{body}} - \beta_1\big(\alpha^* g^*_\theta + \sum_i \alpha_i g^i_\theta\big)$

    Update $\Big(\phi_i \leftarrow \phi_i - \beta_2 g^i_\phi\Big), \Big(\phi' \leftarrow \phi' - \beta_2 g^*_\phi\Big)$

**end**
**Result :** $\theta, \phi'$

---

## A.4 Vision Experiments

We validate that the gains from end-task Aware Training are not siloed to only learning from text. We conduct an experiment comparing end-task aware training on images to its end-task agnostic variant. We use the Cifar100 dataset (Krizhevsky et al., 2009). We use the Medium-Sized Mammals superclass (one of the 20 coarse labels) as our main task whilst the other 19 super classes are used as auxiliary data. Our primary task is thus a 5-way classification task of images different types of

| Method | Medium-Sized Mammals |
|---|---|
| Regular (Task-Agnostic) Pre-training | $46.7_{2.2}$ |
| MT-TARTAN | $51.3_{1.2}$ |
| META-TARTAN | $\mathbf{52.3_{3.8}}$ |

Table 5: We report averages across 3 random seeds. Best average task accuracy is bolded.

medium-sized mammals whilst whilst the remaining 95 classes are grouped into a single auxiliary task.

As can be seen from Table 5, being end-task aware improves over task agnostic pre-training. We find that, again, when our auxiliary task consist of solely domain data and no task data, META-TARTAN performs better than MT-TARTAN (as measured by averaged performance).

## A.5 FULL TAPT TABLE WITH SIGNIFICANCE LEVELS

We repeat Table 1 and provide details about levels of statistical signifance.

| Task | TAPT | MT-TARTAN | $p-$values | META-TARTAN | $p-$values |
|---|---|---|---|---|---|
| ACL-ARC | $67.74_{3.68}$ | $\mathbf{70.48_{4.42}}$ | 0.040 | $70.08_{4.70}$ | 0.069 |
| SCIERC | $79.53_{1.93}$ | $\mathbf{80.81_{0.74}}$ | 0.038 | $\mathbf{81.48_{0.82}}$ | 0.005 |
| CHEMPROT | $82.17_{0.065}$ | $\mathbf{84.29_{0.63}}$ | 0.000 | $\mathbf{84.49_{0.50}}$ | 0.000 |

Table 6: Significance levels as computed from the permutation test. All $p-$values are relative to the TAPT column. Statistically significant performance(p-value from permutation test $< 0.05$), is boldfaced

| Task | TAPT | META-TARTAN | $p-$values |
|---|---|---|---|
| HYPER-PARTISAN | $93.39_{2.26}$ | $\mathbf{96.84_{1.72}}$ | 0.003 |

Table 7: Additional results for HYPERPARTISAN task. This is a binary, partisanship classification task with 515 labeled training examples.

## A.6 FULL DAPT/DAPT+TAPT TABLE

We repeat Table 3 and provide details about levels of statistical signifance.

| Task | DAPT | DAPT+TAPT | MT-TARTAN | $p$-values | META-TARTAN | $p$-values |
|---|---|---|---|---|---|---|
| ACL-ARC | $68.60_{2.62}$ | $69.12_{5.76}$ | $71.58_{1.65}$ | 0.110 | $71.05_{2.37}$ | 0.174 |
| SCIERC | $76.44_{1.19}$ | $77.62_{1.38}$ | $\mathbf{81.02_{1.24}}$ | 0.000 | $\mathbf{81.41_{1.70}}$ | 0.000 |
| CHEMPROT | $80.76_{0.54}$ | $78.22_{0.74}$ | $\mathbf{83.77_{0.60}}$ | 0.000 | $\mathbf{83.38_{0.89}}$ | 0.000 |

Table 8: Duplicate of Table 2. Significance levels as computed from the permutation test. All $p-$values are relative to $\max\big(\mathrm{DAPT}, \mathrm{DAPT} + \mathrm{TAPT}\big)$. Statistically significant performance(p-value from permutation test $< 0.05$), is boldfaced

## A.7 FAQ

1. *What settings are TARTAN algorithms designed for?*
   TARTAN algorithms specialize auxiliary objectives to a particular end-task. This comes at a risk of losing the generic representations afforded by *generalist* pre-trained models. Thus if a practitioner has a sufficiently important end-task where obtaining improved end-task performance is paramount over generic representations, then TARTAN is a viable option.

2. *When do we get computational savings from META-TARTAN?*
   MT-TARTAN does not add any extra overhead compared to pre-train then fine-tune approaches. META-TARTAN however, adds extra overhead per gradient descent step due to computing meta-gradients. However, as shown in Section 5 we are able to get several orders of magnitude improvement in data-efficiency from applying the method. In general,

for the tasks we experimented with, we find that the savings in data-efficiency superseded the extra per-timestep meta-learning overhead.

3. *When should we use META-TARTAN over MT-TARTAN?*

   In +TAPT settings (Tables 1, 3), we observe that META-TARTAN and MT-TARTAN perform similarly. We attribute this to the strength of TAPT-MLM objective. We were pleasantly surprised that the two methods performed comparatively in this setting but in hindsight, we appreciate the insight that went into designing TAPT-MLM as an objective which makes it a strong baseline. In other settings with less carefully designed auxiliary objectives and data (which can potentially be detrimental to the end-task) we expect META-TARTAN to perform better. Section 5.3 provides evidence of this.

