# OpenReview forum: "Should We Be Pre-training? An Argument for End-task Aware Training as an Alternative"
_ICLR.cc/2022/Conference — ICLR 2022 Poster_

### Official Review · Reviewer_nvTM · 2021-11-02

**Correctness:** 3
**Technical Novelty And Significance:** 2
**Empirical Novelty And Significance:** 2
**Recommendation:** 5
**Confidence:** 4

**Main Review:**

The paper is clearly written and well structured.

I believe the question that the authors actually address, whether auxiliary tasks should be used separately or in conjunction with the main task, is important, and their results should be of interest to the community.

However, I think the general framing of their paper in abstract / introduction is misleading. At no point do they train a model from scratch (i.e. without pre-training) with their proposed methods. They do justify this with the high cost of pre-training and the convenient availability of pre-trained models, which ironically would be my main criticisms of actually foregoing generic pre-training. So although they raise the question whether pre-training is necessary, they then don’t actually compare against a model that is not pre-trained. Rather, they show that after pre-training it might not be necessary to further pre-train on large amounts of data just for domain adaptation.

I think the paper would be much stronger if they did not defer their main question (“Should we be pre-training?”) to future work, but rather tested their method with the typical MLM auxiliary task on a newly initialized Transformer model.


**Summary Of The Paper:**

The paper makes the argument that generic pre-training (on auxiliary tasks) is inferior to task-specific pre-training. The authors argue that the final task should be learned together with the auxiliary tasks in a multi-task setup, which they call MT-TARTAN. They also propose a meta-learning algorithm META-TARTAN that uses meta-learning to mitigate the potential impact of updates from auxiliary tasks that detract from the main task.

The authors back this up through experiments. They consider three tasks on CS and Biomed papers, where they train 1) the main task on top of a RoBERTa checkpoint 2) the main task on top of a pre-trained model from Gururangan et al. (2020) and 3) the main task mixed-in with auxiliary tasks from (2) on top of a RoBERTa checkpoint. They observe that their approaches (3) outperform (1) and (2), both in terms of model accuracy and data efficiency. In a setting where the auxiliary tasks are potentially more noisy, they also observe an advantage of META-TARTAN over MT-TARTAN.

Overall their work heavily references Gururangan et al. (2020), who show that adapting a generic model to the task domain through more task-specific pre-training can improve model performance. They make it directly comparable by using their models and a subset of tasks considered in that paper.

**Summary Of The Review:**

I think the paper raises an interesting question, but then only addresses a different, but related question. The results should still be of interest to the community. I think the paper would be much stronger if the authors actually ran experiments that forego pre-training entirely and instead use their mixed-training approach.

---

> ### Author Response · Authors · 2021-11-12
> **Response to Reviewer nvTM**
>
> We thank you for taking time to digest our work. Please find our responses to your specific concerns below.
>
> **Snippet**:  *“However, I think the general framing of their paper in the abstract / introduction is misleading. At no point do they train a model from scratch”*
>
> We reiterate our general response here for specificity. Please note our intention with the title “Should We Be Pre-Training ? An Argument for End-Task Aware Training as an Alternative ” is to inspire further investigation of Pre-training as a whole in the end-task aware setting. We see our progress in the continued pre-training setting (which was chosen because it is computationally tractable for us - and by extension for other practitioners)  as validation that this question has merit and warrants attention. While we did emphasize in the introduction and abstract that we focus on the continued pre-training setting, we recognize that the reviewer found the original text confusing and we have modified the paper to make this clearer.
>
>
> **Snippet**:  *“.... They do justify this with the high cost of pre-training and the convenient availability of pre-trained models, which ironically would be my main criticisms of actually foregoing generic pre-training”*
>
> As we note in the paper,  our prime goal is to improve performance / accuracy on the end-task. We agree with the reviewer that this sacrifices how generic representations of our models are and the ability to amortize the computational cost of leveraging large amounts of data - indeed we do mention this very benefit of generalist models in the introduction. However, as we argue in our general response, for tasks that are sufficiently important to practitioners with targeted applications in mind, this trade-off would be welcome. We leave it to the practitioner to decide which of these objectives is more important to them. Note that for nascent domains looking to leverage machine learning which aren’t mature enough to have access to generalist models - questioning the wisdom of pre-train and finetune as we do here is highly relevant.

---

### Official Review · Reviewer_HAVV · 2021-11-03

**Correctness:** 3
**Technical Novelty And Significance:** 3
**Empirical Novelty And Significance:** 2
**Recommendation:** 6
**Confidence:** 3

**Main Review:**

Strengths:
* The authors advocate for / introduce end-task aware (continue) pre-training, which is a new and practical problem setting for NLP practitioners.
* The authors overcomes the pathological solution problem in their META-TARTAN methods by introducing a separate classification head $\phi^{*}$. Given that meta-learning is known to be unstable and have optimization challenges, the authors' observation and solution is helpful.
* Improved performance (with significance test) and data efficiency compared to DAPT and TAPT proposed in Gururangan et al. 2020.

Weakness:
* The description of the META-TARTAN method is not clear enough.
  * A figure illustrating the data, parameters and optimization steps for META-TARTAN would be very helpful. (e.g., illustrating the relation of $\theta_{body}, \phi^1, \phi^2, ..., \phi^{'}$, how meta-objective is computed using weights $w$ and parameters $\theta$, where does the separate classification head come in, ...)
  * I get to understand the method better (e.g., whether $\phi^*$ is re-initialized at every time stamp $t$, whether $\phi^{'}$ is still being optimized when $\phi^{*}$ appears) only after seeing Algorithm 1 in the appendix. If space allows, please enrich the Sec 3.2-3.3 or move the algorithm to the main text.
* Though the authors put lots of efforts in META-TARTAN, it appears that META-TARTAN is comparable with MT-TARTAN in most cases. META-TARTAN seems to be better in the DAPT-only setting (Sec 5.3); however since we're doing _task-aware_ pre-training, TAPT can be easily achieved in this case. I wonder what is the practical utility of META-TARTAN.
* The paper needs more thorough discussion on computation costs. Computation savings is claimed to be one major advantage of TARTAN. However the comparison is made according to training iterations/steps, while one step in META-TARTAN is much more computationally-expensive than one step in DAPT or TAPT. Please take this into consideration when discussion computation savings.

Questions and further discussion:
* I'm not sure if involving the validation set $D_{T^*}^{val}$ in computing and optimizing the meta-objective is a fair practice. DAPT/TAPT only access it for model selection or early stopping; while META-TARTAN can use it to update meta-parameters $w$ and indirectly influence the training for model parameters $\theta$. Therefore META-TARTAN has more advantage in terms of data usage; this should be discussed in the paper.
* The authors mentioned "an optional fine-tuning step" on the end task in the introduction. Does the results in the paper include this step or not?
* I wonder why in Figure 3 DAPT-MLM is gradually down-weighted, but in Figure 4 DAPT-MLM is up-weighted. Seems to be contradicting. What does Figure 3 look like if we have more iterations?

---
Thanks the authors for their hard work!

**Summary Of The Paper:**

In the paper the authors propose TARTAN, methods to enable end-task aware pre-training. MT-TARTAN simply combines the pre-training objectives and the end-task objective as multi-task learning. META-TARTAN learns a set of weights for the pre-training objective and the end-task objective, using meta-learning. MT-TARTAN and META-TARTAN shows improved performance and data efficiency in a set of three low-resource text classification tasks.

**Summary Of The Review:**

Strength: problem setting is practical and interesting; good performance and improved data-efficiency.
Weakness: lack of visualization and clear description for the method; comparison in computation cost and data usage is questionable.

---

> ### Author Response · Authors · 2021-11-12
> **Response to Reviewer HAVV**
>
> We thank you for taking time to digest our work. Please find our responses to your specific concerns below.
>
> **Snippet**:  *“The description of the META-TARTAN method is not clear enough. … I get to understand the method better … after seeing Algorithm 1”*
>
> We agree that Algorithm 1 makes the paper clearer. We moved the algorithm to the appendix because of space constraints. We will move it back to the main paper in the final version.
>
> **Snippet**: *“Though the authors put lots of effort into META-TARTAN, it appears that META-TARTAN is comparable with MT-TARTAN in most cases … I wonder what is the practical utility of META-TARTAN”*
>
> We repeat a summary of our general response here for specificity. We attribute the comparable performance of *TARTAN methods in the +TAPT settings to the strength of TAPT-MLM objective.  META-TARTAN is designed for two main settings. First, when the auxiliary tasks use OOD data. In this setting we need an algorithm that takes a more nuanced view of the incoming data.  And secondly, when the auxiliary tasks can be detrimental to the end-task. We attribute the fact that META-TARTAN $\approx$ MT-TARTAN without OOD data to the fact that the MLM objectives are really strong auxiliary task baselines and just multitasking them results in a performance boost. It is conceivable that other less established auxiliary tasks which may be detrimental to the end-task will produce a delta in performance between META and MT-TARTAN
>
>
> **Snippet**: *”Computation savings is claimed to be one major advantage of TARTAN. However…. one step in META-TARTAN is much more computationally-expensive than one step in DAPT or TAPT. Please take this into consideration when discussing computation savings.”*
>
> We address this under general concerns to all reviewers but we summarize here for specificity.
> **We agree with reviewers that meta-learning, as with META-TARTAN, adds extra overhead descent per-step, however this overhead is much smaller in comparison to the order of magnitude of the dataset-efficiency we get from applying the method.  We have modified the paper to make this trade-off more explicit.**
>
>
>
> **Snippet**:  *“Questions and further discussion:*
> 1. **Snippet**:  *I'm not sure if involving the validation set D_val…..  more advantage in terms of data usage; this should be discussed in the paper*
>
> Please note that it is standard practice in the meta-learning community to leverage the dev-set in this way [https://arxiv.org/pdf/1911.10088.pdf , https://arxiv.org/pdf/1911.02590.pdf ]. We argue that we are not really using “more data” than other methods since we never train on it directly and we only use it for hyper-parameter selection and model-checkpointing but in a more dynamic way than our baselines do.
>
> 2. **Snippet**:  *The authors mentioned "an optional fine-tuning step" on the end task in the introduction. Does the results in the paper include this step or not?*
>
> Yes, our results include this step. We find that this is sometimes helpful and sometimes not and the improvement is mostly marginal when it is helpful.  We represent each hyper-parameter configuration performance is the max of performance with and without the optional step.
>
> 3. **Snippet**:  *I wonder why in Figure 3 DAPT-MLM is gradually down-weighted, but in Figure 4 DAPT-MLM is up-weighted. Seems to be contradicting. What does Figure 3 look like if we have more iterations?”*
>
> Thank you for the observation. Please note that the presence of TAPT in Figure 4 but not in Figure 3 makes a significant difference. Please note that we address this towards the end of Section 5.4 “.. In general, DAPT is significantly up-weighted compared to the end-task and TAPT. Note that the TAPT + ACL-ARC task weights  (Figure  4) has the same approximate trajectory as the ACL-ARC task weight in Figure 3. It seems important to assign high weight to the task data (Figure 3) but not necessarily all of it needs to go to the actual task loss.(Figure 4)”

---

> > ### Comment · Reviewer_HAVV · 2021-11-30
> > **Reply**
> >
> > Thank you for the clarification and additional results!
> >
> > I agree with other reviewers that the framing of the paper is a little misleading. I see the authors have updated their manuscript. Still, I feel in the current version, the transition from "pre-training" to "continued pre-training" in the abstract is sudden. The reason why a case study on continued pre-training can be linked and extended to general pre-training is not sufficiently justified. Personally I think more revision is needed to make the paper more coherent.
> >
> > On the other hand, I believe this paper presents an interesting problem, and it is a very challenging one. The authors spent a lot of efforts designing TARTAN and overcoming different issues. The work and findings (after revision) will be of interest to the ICLR audience.
> >
> > It's a tough decision for me. My overall score remains the same.

---

### Official Review · Reviewer_ZMKG · 2021-11-04

**Correctness:** 3
**Technical Novelty And Significance:** 2
**Empirical Novelty And Significance:** 3
**Recommendation:** 6
**Confidence:** 4

**Main Review:**

The paper focuses on the continued pre-training setting and it proposes a multi-task end-task aware training method (MT-TARTAN) and a meta-learning variant to achieve better performance than the pre-training + fine-tuning paradigm.

The idea to compare co-training with pre-training-then-fine-tuning is interesting and a promising direction. The proposed method is well motivated to adapt weights during training for the auxiliary tasks and the domain task. The analysis about the tasks weight is interesting in figure 3 and 4, which shows some guidance on how to choose training tasks and dynamically changing the weights during MTL.

There’re a few minor concerns about the paper:

1. Lacking comparisons with other multi-task learning (MTL) work about weight selection
One of the main contributions is to use meta-learning for choosing the weights of each pre-training task. There's a line of work regarding weight selection for MTL. How is meta learning compared to these methods in terms of performance, training speed, etc?

[1] Guo, Michelle et al. “Dynamic Task Prioritization for Multitask Learning.” ECCV (2018).

[2] Kendall, Alex et al. “Multi-task Learning Using Uncertainty to Weigh Losses for Scene Geometry and Semantics.” 2018 IEEE/CVF Conference on Computer Vision and Pattern Recognition (2018): 7482-7491.

2. The setting studied here is kind of limited. Only focusing on low-resource classification tasks. The experiments cover three datasets and show mixed results.
The META-TARTAN doesn’t show much improvement without OOD auxiliary data.
- According to Table 1, the META learning version seems only slightly on two tasks while worse on the other for TAPT. Is there any explanation here?
- It’s more convincing to add more fine-tuning datasets and analyze the effect of dataset size and final performance for the proposed method.

3. One merit about DAPT is that for similar tasks belonging to the same domain, you only need to train DAPT once and then do TAPT. On the other hand, the proposed method always needs to mix in pre-training data to fine-tuning tasks. Given this, TARTAN also introduces overhead and it’s not 100% true that DAPT is more data efficient given that you might only need to go through it once. Not to mention many domains actually have plenty of unlabeled corpus.

A few more questions and comments:
1. What are the number of auxiliary tasks? Seems like for DAPT, TAPT and DAPT+TA, there’s only one auxiliary task? The MLM on either the domain corpus or the target corpus? If that’s the case, then it seems to be too few tasks to be worth selecting from.
2. How hard is it to extend the proposed method to generation tasks instead of discriminative tasks?
3. It’s until the experimental section that I realize the paper focuses on continued pre-training rather than pre-training from scratch. I guess it requires some revision to make that clearer at the beginning of the paper.



**Summary Of The Paper:**

The author argues that direct training on both pre-training task and fine-tuning task would lead to better end-task performance. The key is to incorporate the end task objective function to the pre-training stage.


**Summary Of The Review:**

The MTL f design to combine pre-training tasks and fine-tuning tasks in continued pre-training stage is well motivated. However the experimental setting is a bit limited and the results are also mixed.

---

> ### Author Response · Authors · 2021-11-12
> **Response to Reviewer ZKMG**
>
> We thank you for taking time to digest our work. Please find our responses to your specific concerns below.
>
> **Snippet**:  *“Lacking comparisons with other multi-task learning (MTL) work about weight selection”*
>
> Traditional MTL methods focus on the symmetrical task setting - ie - they seek to improve mean performance across all tasks.
> For example, Guo-et-al learn dynamic task weights based on dynamic performance indicators of each task, thus if 1 task is performing particularly poorly, that task is upweighted to the potential detriment of other tasks even if the other task is our end-task of concern. Similarly in Kendall-et-al, their dynamic weighting seeks to down-weight tasks that have a higher variance / homoscedastic uncertainty. By virtue of the fact that our end-tasks tend to be low resource whilst our auxiliary tasks are high resource,  such a method would tend to down-weight the end-tasks that we care about.
>
> In asymmetrical settings like ours where we care about a particular end-task, these traditional methods could result in trading off good performance on the end-task for better average performance across all the tasks including the auxiliary task. Our method specifically focuses on the end-task to the potential detriment of the auxiliary tasks - as we note in the paper - *“We are willing to sacrifice performance on $T_{\mathrm{aux}}$ if it improves performance on $T^*$ ”*.
>
>
> **Snippet**: *“The setting studied here is kind of limited. Only focusing on low-resource classification tasks. The experiments cover three datasets and show mixed results... It’s more convincing to add more fine-tuning datasets and analyze the effect of dataset size and final performance for the proposed method.”*
>
> We focus our experiments on the low-resource regime because we believe this is the regime practitioners are most likely to boost with auxiliary data. They are also the settings where auxiliary data has been shown to have the most impact [1, 2].
> [1] https://arxiv.org/pdf/2004.10964.pdf
> [2] ​​https://aclanthology.org/2021.acl-long.90.pdf
> We respectfully disagree that our results are mixed, given that when all methods are on an equal footing (in terms of available data) as in Tables 1, 2, 5, we consistently make significant improvements over baselines.
>
> **Snippet:**  *“The META-TARTAN doesn’t show much improvement without OOD auxiliary data. According to Table 1, the META learning version seems only slightly on two tasks while worse on the other for TAPT. Is there any explanation here?”*
>
> We attribute the similar performance of MT and META-TARTAN in settings with TAPT to the fact that TAPT is a strong baseline auxiliary task such that learning dynamic weights in this setting doesn’t result in significant improvement. When we add TAPT to DAPT, both methods perform similarly also - here we suspect that TAPT (also being an MLM objective as with DAPT) provides guidance on which tokens from the OOD data to focus on.  In the pure OOD auxiliary data setting where there is no guidance of the MLM objective from TAPT,  META-TARTAN improves over MT-TARTAN because of its ability to more flexibly adapt to incoming data of variable utility to the end-task.  (please see table 4 in Section 5.3 of the paper)
>
> **Snippet**: *“One merit about DAPT is that for similar tasks belonging to the same domain, you only need to train DAPT once and then do TAPT. … Given this, TARTAN also introduces overhead and it’s not 100% true that …  is more data efficient given that you might only need to go through it once. Not to mention many domains actually have plenty of unlabeled corpus.”*
>
> We agree that performing end task aware training from scratch for each and every end task is daunting. However, for sufficiently important tasks the benefits of improved accuracy of targeted models may outweigh the costs of losing generalization to other tasks. With regard to efficiency, we note that our claims are focused on the setting where training is targeted towards a particular **known end-task** (and not focused towards generalizing to arbitrary new tasks). Here our method reduces the compute burden of continued-pretraining on large amounts of domain data just to achieve improved performance on **solely the end-task**. This setting of a sole important end-task is highly relevant to practitioners who have a **specific application** in mind and are not looking for generic representation models.

---

> > ### Author Response · Authors · 2021-11-12
> > **Continued Response to Reviewer ZKMG**
> >
> >
> >
> > **Snippet**:  *“A few more questions and comments:*
> >
> > 1. *What are the number of auxiliary tasks? Seems like for DAPT, TAPT and DAPT+TA, there’s only one auxiliary task? The MLM on either the domain corpus or the target corpus? If that’s the case, then it seems to be too few tasks to be worth selecting from.*
> >
> > In the DAPT+TAPT experiments, we have 2 auxiliary tasks (MLM with domain data (DAPT) and MLM with task data (TAPT)). Note that **we are not “selecting from two tasks” but we are rather learning how to instantaneously balance / combine the two tasks effectively with the end-task**.  This choice is based on the fact that different tasks might have different relevance to the end-task at different stages in training. Thus instead of just selecting a task and zero-ing out another, we choose to perform the challenging task of balancing them out. Note that our framework can be extended to any number of auxiliary tasks that a practitioner wishes.
> >
> > 2. *How hard is it to extend the proposed method to generation tasks instead of discriminative tasks?*
> >
> > TARTAN works for any end-task with a differentiable objective. It is relatively straight-forward to adapt to generation tasks. The task heads on top of the shared base can be sequence generation models instead of  the classifiers we use.
> >
> > 3. *It’s until the experimental section that I realize the paper focuses on continued pre-training rather than pre-training from scratch. I guess it requires some revision to make that clearer at the beginning of the paper.”*
> >
> > As we mention in the general response, we have updated the early parts of the paper to make this clearer. Please note that in the original version we do mention in the introduction where we list our contributions : **“In lieu of standard end-task agnostic continued pre-training, ….” and in the abstract : “...than the widely-used task-agnostic continued pre-training paradigm of Gururangan et al. (2020)”**.  However, we do agree that this can be made clearer and we have updated the paper to reflect this.

---

### Official Review · Reviewer_uWQC · 2021-11-08

**Correctness:** 4
**Technical Novelty And Significance:** 3
**Empirical Novelty And Significance:** 3
**Recommendation:** 6
**Confidence:** 3

**Main Review:**

Strengths:
1. Formally formularize different configurations of pre-training, finetuning, end-task-aware multitasking or meta learning setups. These formulations are clean and make the difference between the various setup clear.The paper is well-written and easy to understand.
2. The set-up is clean and the experiments are done thoroughly. There are several interesting observations. For example, the observation of task weighting strategies over time from the meta-learning setup is interesting. It shows that in the beginning the end-task is included but with a smaller weight while in the later phase the end-task is upweighted but the auxiliary tasks still receives small weights, instead of being all zeros.

Weaknesses:
1. Intuitively, it is not surprising that the proposed method performs better than other end-task agnostic pretraining. While the paper argues that the end-task aware setting, when done correctly, will save computational power, it does not consider the comparisons when there are multiple end-tasks involved. It is understood that a shared pretrained model has the advantage of quick iterations on various end-tasks. When there are a large number of end-tasks involved, it become daunting to train/pre-train large models for each end-task.

Though I also acknowledge that the end-task aware approach proposed by the paper is a good middle ground when there is a specific end task and medium amount of resources are available, which makes training useful large models more accessible, since the end-task aware method are more data-efficient (resource-efficient.)

**Summary Of The Paper:**

This paper considers the common setup of pretraining and finetuning paradigm and argues that an end-task aware setting, either by multitasking or an additional meta learning that learns the weights between various auxiliary tasks and the end task, is superior than end-task agnostic pretraining.

The paper provides formal formulations of the various set-ups. Propose an interesting strategy for the meta-learning setup (Section 3.3).

**Summary Of The Review:**

The paper provides supporting evidence that end-task aware training can achieve a better performance while being more data-efficient. The paper is well-written and easy to follow. It provides an alternative to the pretraining + finetuning setup and their observations that end-task aware approaches are more data-efficient might make training large models more accessible to entities with slightly less computational resources.

---

> ### Author Response · Authors · 2021-11-12
> **Response to Reviewer uWQC**
>
> We thank you for taking time to digest our work. Please find our responses to your specific concerns below.
>
> **Snippet** : *“Intuitively, it is not surprising that the proposed method performs better than other end-task agnostic pretraining”*
>
> Given that end-task aware training is rarely used (we don't know any prior work)  as an alternative to classical pre-training in practice, we argue that this is not simply validating already-known information. We argue that there is a place in the literature for demonstrating the effectiveness of intuitive methods that are reasonable in hindsight, but not standard practice.
>
> **Snippet**: *“”When there are a large number of end-tasks involved, it become daunting to train/pre-train large models for each end-task”*
>
> We agree that performing end task aware training from scratch for each and every end task is daunting. However, for sufficiently important tasks the benefits may outweigh the costs. In addition, given the positive results in our paper, we see much potential for future work into methods that achieve similar improvements while reducing compute requirements, such as applying TARTAN to multiple end tasks simultaneously, or further exploring combinations of TARTAN with continued pre-training.

---

### Author Response · Authors · 2021-11-12
**General Response**

We would like to thank all reviewers for dedicating time to read our work. We appreciate your feedback and its contribution towards making our paper stronger. We address some of your common concerns below

**[Clearer Title, Abstract and Introduction]**

Please note our intention with the title “Should We Be Pre-Training ? An Argument for End-Task Aware Training as an Alternative ” is to inspire further investigation of Pre-training as a whole in the end-task aware setting. We see our progress in the continued pre-training setting (which was chosen because it is computationally tractable for us - and by extension for other practitioners)  as validation that this question has merit and warrants attention.
While we did emphasize in the introduction and abstract that we focus on the continued pre-training setting, we recognize that multiple reviewers found the reference to Pre-training in the title to be misleading and did not realize sufficiently early that we are focused on the continued-pretraining setting. **We have modified the title, abstract and introduction to make it clearer that we operate in the continued pre-training setting**.


**[Comparison of Computational Savings]**

We agree that performing end task aware training from scratch for each and every end task is daunting. However, for sufficiently important tasks the benefits of improved accuracy of targeted models may outweigh the costs of losing generalization to other tasks. With regard to efficiency, we note that our claims are focused on the setting where training is targeted towards a particular known **end-task** (and not focused towards generalizing to arbitrary new tasks) - a highly relevant setting to practitioners who have a **specific application** in mind and are not looking for generic representation models.

**We agree with reviewers that meta-learning, as with META-TARTAN, adds extra overhead per descent step, however this overhead is much smaller in comparison to the order of magnitude of the dataset-efficiency we get from applying the method.  We have modified the paper to make this more explicit**.

Also, given the positive results in our paper, we see much potential for future work into methods that achieve similar improvements while reducing compute requirements, such as applying TARTAN to multiple end tasks simultaneously, or further exploring combinations of TARTAN with continued pre-training


**[Benefit of META-TARTAN over MT-TARTAN]**

We attribute the comparable performance of *TARTAN methods in the +TAPT settings to the strength of TAPT-MLM objective. We were pleasantly surprised that the two methods performed comparatively in this setting but in hindsight, we appreciate the insight that went into designing TAPT-MLM as an objective which makes it a strong baseline. In other settings with less carefully designed  auxiliary objectives and data (which can potentially be detrimental to the end-task) we expect META-TARTAN to perform better. Section 5.3 provides evidence of this.

---

### Author Response · Authors · 2021-11-29
**Adding Extra Experimental Datapoint**

Dear Reviewers, thanks once again for your feedback. We hope that our responses helped assuage any concerns, and would be happy to respond to any additional ones as necessary. We would also like to present the following TARTAN results on a new dataset+domain for your consideration as you finalize your ratings for our paper.
We present this experiment mainly in response to reviewers that expressed that using 3 tasks was a limited setting. We hope to further strengthen our method by expand to more regimes in the final version.

| DOMAIN 	| TASK          	| ROBERTA        	| TAPT           	| META-TARTAN    	|
|--------	|---------------	|----------------	|----------------	|----------------	|
| NEWS   	| HYPERPARTISAN 	| $93.39_{2.26} $	| $93.42_{2.87}$ 	| $\mathbf{96.84_{1.72}}$	|

---

### Decision · Program_Chairs · 2022-01-20

**Decision:**

Accept (Poster)

**Comment:**

The authors argue in favor of task-aware continued pretraining and demonstrate through experiments that using objectives based on the end-task during continued pretraining help in improving downstream performance.

The reviewers generally appreciated the motivation, the formal treatment of the topic and the thoroughness in the experiments. There were some concerns about (i) positioning of the paper (pretraining as opposed to continued pre-training) (ii) thorough comparison with other MTL frameworks (iii) evaluating on more datasets (iv) cost of continued pretraining for each task v) the benefit of META-TARTAN over MT-TARTAN only in specific  settings and (vi) lack of surprise/novelty in the results.

IMO, the authors have adequately addressed ALL the above concerns raised by the reviewers. Further, despite the above concerns, all reviewers agree that the problem is well motivated and of interest to the community and most aspects of this work are thorough. The findings will be useful and may spawn other work in this area.